

# Characteristics of wintertime VOCs in suburban and urban Beijing: concentrations, emission ratios, and festival effects

Kun Li[1,2,3], Junling Li[1,2], Shengrui Tong[1,2], Weigang Wang[1,2], Ru-Jin Huang[4], and Maofa Ge[1,2,5]

[1]Beijing National Laboratory for Molecular Sciences (BNLMS), State Key Laboratory for Structural Chemistry of Unstable and Stable Species, CAS Research/Education Center for Excellence in Molecular Sciences, Institute of Chemistry, Chinese Academy of Sciences, Beijing 100190, China

[2]University of Chinese Academy of Sciences, Beijing 100049, China.

[3]Air Quality Research Division, Environment and Climate Change Canada, Toronto, Ontario M3H 5T4, Canada.

[4]Key Laboratory of Aerosol Chemistry & Physics, State Key Laboratory of Loess and Quaternary Geology, Institute of Earth Environment, Chinese Academy of Sciences, Xi'an 710061, China

[5]Center for Excellence in Regional Atmospheric Environment, Institute of Urban Environment, Chinese Academy of Sciences, Xiamen 361021, China

Correspondence: Maofa Ge (gemaofa@iccas.ac.cn)

**Abstract.** Measurements of volatile organic compounds (VOCs) were performed at a suburban site and an urban site in Beijing during the winter of 2014-2015. The VOC concentrations and emission ratios (ERs) to CO were compared at these two sites. It is found that though the VOC concentrations at the urban site are 2.67±1.15 times of those at the suburban site, the ERs are similar (within a factor of 1.5). It is indicated that: 1. the VOCs at suburban areas are mainly from the transportation from the urban areas; 2. the ERs measured at the urban areas are also valid for the surrounding suburban areas.

By comparing the diurnal variations and the contribution of anthropogenic emissions at both sites, we find that the photochemical processes are very active at the urban site, and these processes play an important role in the daytime oxygenated VOCs (OVOCs) formation. The methanol at the urban site and the formic acid at the suburban site probably have additional sources, which are attributed to be solvent use and soil/agriculture, respectively. The festival effects from Chinese New Year (CNY) were investigated. The VOC concentrations decreased ~60% during CNY holidays, probably due

to the population migration during festival holidays. In addition, fireworks are found to be an important source of acetonitrile, aromatics, and some OVOCs during CNY festival, and should be controlled more strictly. This study provides key characteristics of wintertime VOCs in suburban and urban Beijing, and has implications for better understanding the atmospheric chemistry of VOCs in and around this megacity.

# 1 Introduction



Volatile organic compounds (VOCs) play important roles in air quality as they can form secondary pollutants such as ozone and secondary organic aerosol (SOA) during their oxidation processes in the atmosphere (Atkinson and Arey, 2003;Seinfeld and Pandis, 2006). Atmospheric VOCs have diverse primary emission sources including both anthropogenic and biogenic sources. While biogenic emissions dominate over anthropogenic emissions on a global scale (Guenther et al., 2006;Park et

al., 2013), anthropogenic emissions play a more important role in urban and surrounding areas (Warneke et al., 2007;de Gouw et al., 2009;Ait-Helal et al., 2014;Wu and Xie, 2018). Besides primary emissions, some oxygenated VOCs (OVOCs), such as aldehydes, ketones, and carboxylic acids, can be formed in the atmosphere through oxidation processes (Chen et al., 2014;Millet et al., 2015;Friedman et al., 2017), which makes their sources more complex. Separation of primary emission and secondary formation contributions of OVOCs remains challenging, especially for megacities with various anthropogenic

emissions and chemical processes (Parrish et al., 2012;Yuan et al., 2012).

As anthropogenic emissions dominate the VOCs sources in megacities and surrounding areas, accurate estimation of the amount of anthropogenic emissions is of great significance for better understanding the role of VOCs in atmospheric chemical processes. In general, there are two approaches to estimate the anthropogenic emissions: "bottom-up" method and "top-down" method (Borbon et al., 2013;Wang et al., 2014). The bottom-up approach is achieved by summing up emissions

from all known individual activities, while the top-down approach is a measurement-based method providing emission rates of individual compounds. The top-down approach is usually more accurate and is used to evaluate the accuracy of bottom-up emission inventory (Wang et al., 2014;Li et al., 2017c). Using the VOC emission ratios (ERs) relative to inert species, e.g. carbon monoxide (CO) is a good top-down approach. CO has good correlations with most anthropogenic VOCs, and the emission of CO is simple and well understood (Warneke et al., 2007;Bon et al., 2011;Borbon et al., 2013;Wang et al., 2014).

Hence, characterization of these emission ratios is essential for better understanding the anthropogenic emissions of VOCs in megacities and to formulate effective control policies.

Beijing, the capital and one of the largest megacities in China, has suffered from severe air pollution in the past few decades (Chan and Yao, 2008;Zhang et al., 2009;Huang et al., 2014;Fu and Chen, 2017). Understanding the characteristics of VOCs (such as concentrations, diurnal variations, and emission features) in urban and suburban Beijing has great significance for

understanding their atmospheric chemistry and mitigating these air pollutants. The severe haze pollution in Beijing often occurs during winter (Sun et al., 2013;Zhang et al., 2015;Zheng et al., 2015;Elser et al., 2016). However, the data of wintertime VOCs at urban and/or suburban Beijing is still very scarce (Guo et al., 2017;Liu et al., 2017), leading to difficulties in quantifying the VOC levels and emissions in Beijing and surrounding areas. In addition, during Chinese New Year (CNY), two factors may dramatically influence the VOC concentrations and emission features in Beijing: population

migration and fireworks. First, the population in Beijing drastically decreased by more than 50% and most industrial activities were closed during CNY, because many people in this megacity move back to their hometown for the festival. Hence, CNY is a good time to investigate the effects of human activities on the pollutant levels and emission features of megacities. Second, the fireworks and firecrackers during CNY have an important impact on the air quality (Zhang et al.,



2010;Li et al., 2013;Cheng et al., 2014;Jiang et al., 2015;Li et al., 2017a). However, most of the previous studies focus on aerosols from fireworks, leaving the VOC emissions from CNY fireworks poorly understood.

In this study, we conducted on-line measurements of 14 VOC species at suburban and urban Beijing during the winter of 2014-2015. The measured VOCs include both hydrocarbons and OVOCs. The VOC concentrations at suburban and urban sites are reported and compared with each other and with previous studies. The diurnal patterns of VOCs at both sites are shown, from which the effects of primary emissions and photochemical processes on individual species are discussed. The emission ratios of these VOCs to CO at both sites are estimated. Using these ERs, the contribution of primary anthropogenic emission to each OVOC is estimated. In addition, the effects of human activities during Chinese New Year holidays on VOC emissions are investigated. The aim of this study is to better understanding the characteristics of wintertime VOCs at urban and suburban Beijing, especially the VOC concentrations, diurnal variations, emission features and the effects of human activities.

## 2 Methods

### 2.1 Sampling sites and sampling time

The measurements were conducted at a suburban site and an urban site (Fig. 1) during the winter of 2014-2015. The suburban site (40°24'30'' N, 116°40'29'' E) is on the 4th floor of the teaching building #1 at University of Chinese Academy of Sciences (UCAS), which is about 50 km away from downtown (the North 5th Ring Road). This site has been described in detail in a previous study (Li et al., 2017b). Briefly, this is a typical suburban site, with a residential area about 500 m away at the northeast and a road with sparse traffic about 100 m away at the east. The sampling period at this site is from November 24th to December 24th, 2014. The urban site (39°59'12'' N, 116°19'06'' E) is on the rooftop of a five-story building at National Center for Nanoscience and Technology of China (NCNST). This site is close to the North 4th Ring Road (~200 m), a road with heavy traffic. The sampling time at this site is from January 31st to March 1st, 2015. This sampling time is divided into two periods: the first period is from January 31st to February 15th, which is the normal days; the second period is from February 17th to 28th, which is roughly the holidays of Chinese New Year (CNY). The two periods are shown in Fig. 2. Based on the concentrations of CO, SO$_2$, NO$_x$, and VOCs, we conclude that February 16th and March 1st are the transition time between these two conditions; hence they are not included in both periods.

### 2.2 Measurements

VOC concentrations were on-line measured with a quadrupole proton transfer reaction-mass spectrometry (PTR-MS, Ionicon Analytik) (Lindinger et al., 1998). The CO, SO$_2$, NO$_x$ concentrations were measured by corresponding gas analyzers (Ecotech). The operation principle and the deployment of the PTR-MS have been described in a previous study (Li et al., 2017b). Briefly, VOC molecules react with hydronium ions (H$_3$O$^+$) in a drift tube reactor and generate VOC·H$^+$ ions (i.e.: protonation). These ions are then selected by a quadrupole mass filter and detected by an electron multiplier. Using PTR-



MS, only the species with a proton affinity greater than $H_2O$ (691 kJ $mol^{-1}$) can be protonated and detected. During the measurements, the pressure of the drift tube was maintained at 2.2 mbar, and the reduced electric field parameter (E/N, where E is the electric field and N is the gas number density) was 130 Td. The temperatures of the inlet line and the drift tube were both kept at 60 ˚C. The time resolution of the PTR-MS is 15 s. The sampling flow of the PTR-MS is about 110

5    mL $min^{-1}$, and a side pump with a flow rate of 5.5 L $min^{-1}$ was used for sampling.

The measured VOC·$H^+$ ions and corresponding VOC species are listed below: 1. m/z 42 (acetonitrile); 2. hydrocarbons including m/z 69 (isoprene), m/z 137 (monoterpenes), m/z 79 (benzene), m/z 93 (toluene), m/z 105 (styrene) and m/z 107 (C8 aromatics, including ethylbenzene and xylenes); 3. oxygenated VOCs (OVOCs) including m/z 33 (methanol), m/z 45 (acetaldehyde), m/z 47 (formic acid), m/z 59 (acetone), m/z 61 (acetic acid), m/z 71 (methyl vinyl ketone and methacrolein,

MVK+MACR,), and m/z 73 (methyl ethyl ketone, MEK). Two background measurements were performed daily by sampling air through a Supelpure hydrocarbon trap (Supelco). The data are processed with the PTR-MS Viewer software (Version 3.1). The raw counts are normalized by the signal of the $H_3O^+$ isotope, m/z 21, and are corrected by background subtraction. Although the PTR-QMS only detects the integer masses, previous comparison studies showed that most of these ions were influenced little by other species in the atmosphere (de Gouw and Warneke, 2007;Yuan et al., 2017). Possible

interferences are discussed in the Results section.

The PTR-MS was calibrated with a dynamic calibrator (Thermo 146i) and a standard gas cylinder containing 65 VOCs with a mixing ratio of 1 ppm for each species (TO 15, Linde). The calibration factors were found to be within ±5% before and after each campaign. Ten measured VOC species are in the calibration gas cylinder except acetonitrile, formic acid, acetic acid, and monoterpenes. For these four species, the transmission curve (which was calibrated using the ten species in the

calibration gas) and the reaction rates with $H_3O^+$ were used to calculate the concentration (Taipale et al., 2008;Zhao and Zhang, 2004). The sensitivities calculated by this method were compared with the permeation tubes of these four species (VICI), and the differences were less than 20%. As 20% is the uncertainty of the permeation tube, so we use the calculated sensitivities of the four species in this study.

## 3 Results and discussions

### 3.1 Urban and suburban VOC concentrations

The VOC concentrations at the urban and suburban sites are shown in Table 1 and illustrated in Fig. 3. The VOC concentrations in autumn 2014 of both urban and suburban site (Li et al., 2015;Li et al., 2017b) are also shown in Fig. 3 for comparison. At the suburban site, all VOC concentrations measured in this study are slightly lower compared to the VOC concentrations in autumn 2014 (Li et al., 2017b). The possible explanation for this decrease is the change of meteorological

factors. The weaker solar radiation of winter compared with autumn can cause the less formation of biogenic VOCs and secondary OVOCs; the decreased temperature (about 10 °C lower, Fig. S1) leads to the less emission of hydrocarbons from evaporation, such as gasoline evaporation, painting, and printing. For the urban site, we use only the Period I data to avoid



any influence from CNY holidays (Fig. 2). Compared with the concentrations at Peking University (PKU) during autumn 2014 (Li et al., 2015), the OVOC concentrations of this study are similar, but the concentrations of acetonitrile, toluene, C8 aromatics, and isoprene are lower. As these species are all main component of biomass burning emissions (Warneke et al., 2011;Li et al., 2017b), the decrease of these species may be a result of the decrease of biomass burning emissions from autumn to winter.

As shown in Fig. 3, the concentrations of all VOCs at the urban site are higher than those at the suburban site. The ratios of urban to suburban average concentrations for each compound are marked in Fig. 3 and summarized in Fig. 4. The average ratio is 2.67±1.15. The concentrations of MEK and isoprene at the urban site are about 5 times of those at the suburban site, which are the highest factors. As will be discussed in the next section, the difference in MEK is likely caused by the secondary source, while the differences in isoprene may have several possible explanations. The concentration of methanol at the suburban site is 3.8 times of that at the suburban site, and this difference is probably from the solvent use from chemistry research institutes close to the urban site, which will be discussed in Section 3.3. As listed in Table 1, the VOC concentrations at the UCAS site are basically higher than those at Changdao (Yuan et al., 2013), a typical rural site, indicating that the VOC levels at UCAS are influenced by transportations from urban areas, such as urban Beijing and the city groups at Beijing-Tianjin-Hebei area (Li et al., 2017b).

### 3.2 Diurnal variations

The diurnal variations of VOCs at urban and suburban sites are shown in Fig. 5. As acetonitrile reacts very slowly with OH ($k_{OH}$ = 2.2×10$^{-14}$ cm$^3$ molecule$^{-1}$ s$^{-1}$) (Atkinson et al., 2006), it is usually considered to be a good tracer for primary emissions. As can be seen from Fig. 5a, the concentration of acetonitrile decreases at daytime, which is due to the change of boundary layer height (de Gouw et al., 2009;Bon et al., 2011;Yuan et al., 2012). The peaks of acetonitrile at 8:00-10:00 local time (LT) are due to the pollutants accumulation with a shallow boundary layer height. The patterns of acetonitrile at both sites are considered to be combinations of primary emissions and variation of boundary layer height.

As shown in Fig. 5a, the hydrocarbons at the urban site have several minor peaks (e.g. 2:00 and 18:30 LT) that are not observed at the suburban site. These peaks are suggested to be vehicle emissions. The 2:00 LT peak is mainly from the freight trucks as they are not allowed to enter urban areas of Beijing before 0:00 LT and during daytime; the 18:30 LT peak is most likely from traffic rush-hour emissions. At the suburban site, the daytime decrease ratios of all hydrocarbons are higher than acetonitrile, which is due to their higher reaction rate constants with OH. At the urban site, some hydrocarbons, such as aromatics, have lower daytime decrease compared with acetonitrile, which is likely due to strong local emissions. The daytime reduction of urban isoprene is much lower than other VOCs, and also lower than suburban isoprene, which may be caused by the following reasons. First, signals at m/z 69 have interferences from furan (de Gouw and Warneke, 2007), which is mainly from anthropogenic sources such as the combustion of fossil fuels and waste. Hence, the furan interference may be higher in urban areas. Second, the plant species are different at urban and suburban sites. For example, some broadleaf trees and shrubs can generate more isoprene than others (Guenther et al., 2006;Sharkey et al., 2008). The tree



species are artificially selected at urban areas, while tree species at suburban areas are more natural. Third, there are probably some anthropogenic sources of isoprene, such as motor vehicles (Borbon et al., 2001;Barletta et al., 2002;Li et al., 2017b).

As shown in Fig. 5b, OVOCs have additional sources at daytime compared to inert species (e.g. acetonitrile); they are mainly attributed to secondary productions. The daytime OVOC levels are lower than nighttime at the suburban site.

However, at the urban site, the secondary productions of OVOCs are so strong that they almost offset the daytime "valley" caused by boundary layer height variations. This indicates that the photochemical reactions are more active at the urban site. The diurnal variations of suburban formic acid are quite different from other suburban OVOCs, with little change during the whole day. We speculate that there are large formic acid sources in suburban areas, such as soil and agriculture (Sanhueza and Andreae, 1991;Millet et al., 2015). There is a remarkable enhancement in MVK+MACR concentration during daytime

compared with the pattern of acetonitrile at the urban site, which is not observed at the suburban site. As the oxidation of isoprene is the main source of MVK+MACR, we suggest that the observed isoprene enhancement in Fig. 5a at the urban site is not totally from the furan interference. In other words, there are indeed more daytime isoprene emissions at the urban site compared to the suburban site. MEK is the OVOC with the most different concentrations at suburban and urban sites (Fig. 4), and it has complex atmospheric sources including biogenic, anthropogenic and secondary sources (Yáñez-Serrano et al.,

2016). It is shown in Fig. 5b that the urban MEK has no obvious peak at rush hours. Instead, the MEK concentration follows the variations of solar radiation, indicating its biogenic/secondary sources. It is more likely from secondary sources, as the urban average MEK concentration (1.89 ppb) is much higher than that of typical biogenic emission dominant sites, which is typically < 0.5 ppb (Yáñez-Serrano et al., 2016). Suburban MEK also shows a peak at 14:00-15:00 LT, but much lower than that at the urban site.

To better shown the difference in VOCs emissions at urban and suburban sites, the diurnal variations of differences in VOC concentrations are shown in Fig. 6. Clearly, there are three rush-hour peaks from vehicle emission at about 2:00, 7:40 and 18:30 LT, respectively. Two of these peaks have been shown in Fig. 5a. From these three peaks, we know that there is a lot of VOC species related to vehicle emission. First, all aromatics show obvious peaks at rush hours, indicating that vehicle emission is their main source. Second, acetonitrile has minor peaks at rush hours, indicating that vehicle emission is one of

the sources, but there are still other primary sources (e.g. biomass burning). Third, vehicle emission is a source of some OVOCs such as acetone, acetaldehyde, formic acid, and acetic acid. At last, vehicle emission is also a contributor of isoprene and monoterpenes. However, the broad peak from 10:00 to 16:00 LT indicates that the urban biogenic emission of isoprene and monoterpenes are also stronger than the suburban site. The most obvious secondary production at the urban site is from acetone and MEK. They both have a broad peak from 12:00 to 17:00 LT, which is most likely from photooxidation. There

are probably secondary productions of other OVOCs as well, though they are not obviously shown in the diurnal variations of Fig. 6. The contribution of the secondary source on the production of OVOCs will be discussed in Section 3.4.

**3.3 Emission ratios**





VOC emission ratios to inert species such as CO are important parameters that can be used to quantify anthropogenic emissions. Generally, there are two approaches to estimate the emission ratios of VOCs to CO (Borbon et al., 2013). The first one is the photochemical-age-based method, which is described in details by de Gouw et al. (2005) and Warneke et al. (2007). However, this method is quite dependent on the accurate estimation of the ratio of a VOC pair when these species

are freshly emitted, and the errors of this ratio will lead to deviations in ERs (Warneke et al., 2007). The second one is the linear regression fit of the VOC-CO scattering plot. The key to this method is to prevent photochemical influence on the regression. In this study, we use the linear regression method because: 1. it is more commonly used and the photochemical influence can be avoided by using nighttime dataset (Bon et al., 2011;Borbon et al., 2013); 2. the estimation of the ratio of a freshly emitted VOC pair may have large uncertainties due to the limited choices of VOC species measured in this study.

The ERs for both urban and suburban sites are estimated by the linear orthogonal distance regressions (ODR) (Bon et al., 2011;Wang et al., 2014) using the data of 0:00-4:00 LT, and are listed in Table 2. ERs from some related studies are also shown for comparison. Though only the nighttime data is used for the regression, it is shown in previous studies that the daytime ERs are very similar to nighttime ERs (Bon et al., 2011;Borbon et al., 2013). To verify this, we plot two inert primary VOCs, acetonitrile and benzene, versus CO in Fig. S2 using all day data and data of 0:00-4:00 LT, respectively.

Both acetonitrile and benzene are mainly from anthropogenic emissions and react slowly with OH, hence minimize the influence of consumption and production from the reactions with OH at the day. As shown in Fig. S2, the ERs retrieved from 0:00-4:00 LT data are still valid when extended to all day data.

It is shown in Fig. 7a that the ERs between urban and suburban site agree very well, mostly within a factor of 1.5. The exceptions are formic acid and methanol, which have higher emission ratios at the urban site. The higher emission ratios of

formic acid and methanol may be a result of the influence of solvent use from chemistry research institutes close to the sampling site. There are three chemistry institutes within the distance of 1 km from the sampling site that may influence the signal of formic acid (from the interference of ethanol) and methanol: NCNST itself; Institute of Chemistry, Chinese Academy of Sciences; College of Chemistry and Molecular Engineering, Peking University. The high concentrations of ethanol and methanol from the plumes of these institutes may enhance the measured ERs of formic acid and methanol.

Because the sensitivity of PTR-MS to ethanol is much lower than formic acid (Yuan et al., 2017), the influence of ethanol on formic acid is relatively minor, which could be proved by its high linear relevance to CO (R = 0.85). However, the solvent may greatly influence the ER of methanol, as the linear relevance to CO is very low at the urban site (R = 0.16), but high at the suburban site (R = 0.80). The previous studies conducted at the PKU site (Yuan et al., 2012;Wang et al., 2014) may also be influenced by solvent use, as the PKU site is close to the NCNST site.

The comparison of this study to the wintertime ERs of a previous study conducted at urban Beijing (Wang et al., 2014) is shown in Fig. 7b. The ERs at NCNST and UCAS generally agree with those at PKU site (within a factor of 2). Exceptions are isoprene and methanol of both sites. The methanol ER at PKU is between the urban and suburban ERs of this study, which is speculated to be a result of lower influence from solvent use compared with NCNST. The comparison of this study to a previous study conducted at a rural site (Changdao, Fig. 7c) shows larger discrepancies. ERs of acetonitrile and



monoterpenes are higher at Changdao (Yuan et al., 2013), which could be a result of the higher contribution of biomass burning at this site compared with Beijing. The higher ER of methanol at Changdao compared with UCAS might also associate with enhanced biomass burning at the rural site. The higher ERs of isoprene at the two sites of this study may be a result of more vehicles at Beijing and surrounding areas. When comparing these ERs with the data of summer Beijing (Yuan et al., 2012), the discrepancy is even larger. As shown in Fig. 7d, the plots are mostly below the 1:1 line, which means that ERs of summertime urban Beijing are larger than ERs of both sites of this study. For most of the VOCs, the summertime ERs are 2-4 times higher than those of wintertime, indicating a large seasonal difference in emissions. The same level of methanol ERs at summer and winter indicates a relative constant emission from chemical use.

## 3.4 Anthropogenic and biogenic/secondary contribution to OVOCs

The sources of OVOCs are very complex; hence understanding their anthropogenic and biogenic/secondary contributions is of great significance. Using the emission ratios above and the concentrations of CO, one can estimate the primary anthropogenic emission of individual OVOC species. The amount of other sources (mainly biogenic/secondary) for each OVOC can be estimated by:

$$Biogenic/secondary = VMR - PA = VMR - ER \times CO \qquad (1)$$

where PA and VMR are the primary anthropogenic component and the total volume mixing ratio of a given VOC (Brito et al., 2015;Sheng et al., 2018). As discussed above, the ERs derived from nighttime data are still applicable during the whole day.

The contributions of anthropogenic and biogenic/secondary of OVOC species at the urban site is shown in Fig. 8a. During the day, the contribution of biogenic/secondary (mostly secondary) increased significantly for all OVOCs, and the maximum secondary contribution appeared at 14:00-16:00 LT. The maximum secondary contribution of MEK is the highest, which is nearly 90%. The second highest is formic acid (~75%) and acetone (~70%). The high secondary contributions of MEK and acetone agree well with the diurnal variations shown in Fig. 6. The maximum secondary contribution of acetaldehyde, acetic acid, and MVK+MACR is 50% – 60%. These high secondary contributions indicate the strong photochemical process during the day at the urban site, despite the low temperature (about 0 – 10 °C) during the sampling period. The high secondary contributions of OVOCs agree well with previous studies during Beijing winter. For example, Chen et al. (2014) found that the secondary contribution of carbonyls during winter is 51.2%, which is similar to the secondary contribution during summer (46%).

As shown in Fig. 8b, the local biogenic/secondary emissions of formic acid contribute ~80% of the total sources at the suburban site, which verifies the assumption in Section 3.2. For other OVOCs, the anthropogenic emissions dominate the sources at the suburban site. There are two possible explanations for this. First, the suburban VOC and oxidant (such as OH, $O_3$) concentrations are much lower than the urban site, leading to the weak secondary production of OVOCs, which is shown in the diurnal trends in Fig. 5b. Second, regional transportation plays a more important role than local emissions at the



suburban site. During transportation, oxidation processes produce some secondary OVOCs. However, as these secondary OVOCs arrive at the suburban site at the same time with CO during transport, they are classified as "primary anthropogenic". Hence, we conclude that it should be careful when applying this method to sites with weak local emission and strong regional transportation.

## 3.5 CNY festival effects

### 3.5.1 Concentrations and emission ratios

The population migration during CNY holiday is large. According to the data from the Beijing Municipal Bureau of Statistics (http://www.bjstats.gov.cn/tjsj/), the resident population in Beijing was 21 billion, and the floating population was about 8 billion at 2014-2015. During CNY holidays, about 40% of the resident population and more than 90% of the floating population left Beijing. In other words, the Beijing population during CNY was about 45% (40% – 50%) of normal time. The effects of this large amount of population migration on VOC concentrations and ERs are discussed in this section.

The comparison of average concentrations at the urban site during the non-CNY time (Period I) and CNY holidays (Period II) are shown in Fig. 9a. It is shown that the concentrations of all VOC species during the non-CNY period are higher than those during the CNY period, with an average ratio of 2.5±0.63. In other words, the VOC concentrations during CNY holidays are only 40% (32% – 53%) of those during the non-CNY period. This percentage is similar to the percentage of the population that stay in Beijing during CNY, which is about 40% – 50%. As shown in Fig. 2, the NO concentration decreased drastically during CNY holidays, which is likely a result of the reduction of vehicles. The severe decline of aromatics, acetonitrile, and isoprene may also from vehicle reduction. The $SO_2$ concentration decreased as well (except several peaks from fireworks), indicating the emissions from coal burning also decreased during CNY holidays. The emission reduction from vehicles and coal burning are all related to the population migration. In addition, other population-related anthropogenic emissions would decrease as well, such as cooking, solvent, and construction. The drastically decrease of methanol is very likely a result of reduced solvent use. Another interesting finding is that the VOC concentrations during CNY holidays at the urban site are very similar to those of the suburban site (Fig. 9b). The average ratio of concentrations of urban Period II to suburban is 1.05±0.26. It indicates that the population migration during CNY holidays can reduce the urban VOC concentrations to suburban levels.

Table 3 shows the VOC emission ratios at the CNY period and their change percentage compared with the non-CNY period. 11 out of the 14 ERs decrease, and most of the emission ratios changed for less than 30%, except methanol. It indicates that the VOCs emission features didn't change too much, though the emission intensity decreased drastically. The dramatical decrease of methanol ER (86%) indicates its strong emission during the non-holiday period, which is very likely to be solvent use.

### 3.5.2 Fireworks





As shown in Fig. 2, the $SO_2$ concentration increased drastically at February $18^{th}$ – $19^{th}$ (the CNY eve and festival), which was likely caused by fireworks (Chang et al., 2011). The time series of representative VOCs, CO, and $SO_2$ from February $18^{th}$ 10:00 to February $19^{th}$ 18:00 are shown in Fig. 10. According to the $SO_2$ concentration, there were three fireworks episodes at night of February $18^{th}$ (F1), and midnight (F2) and morning (F3) of February $19^{th}$, respectively. Among these episodes, F1 and F3 are similar: VOCs and CO increased with the increasing $SO_2$. F2 is much different, with a high $SO_2$ peak but no obvious change in VOCs and CO. F2 is at 0:00 of February $19^{th}$, which is the most important moment of the CNY festival celebration. Hence, the fireworks displayed at this time mostly have high energy (e.g. display shells). Though this kind of high-energy firework emission contains high concentration of $SO_2$ and aerosols, the VOCs emission is very low according to a recent study (Xu et al., 2018). However, during F1 and F3, more low-energy fireworks (e.g. firecrackers) were used. The use of low-energy fireworks can emit a large amount of VOCs such as aromatic hydrocarbons and phenols (Xu et al., 2018), hence the VOCs enhancements were observed during F1 and F3. The different emission features of high-energy and low-energy fireworks are likely due to the difference in combustion efficiency. The VOCs species from fireworks reported in previous studies also have large differences. For example, Chang et al. (2011) found that the benzene, toluene, ethylbenzene, and xylenes (BTEX) concentrations were all increased during a firework episode in Taiwan, China. However, Drewnick et al. (2006) found that the concentrations of methanol, acetonitrile, acetone, and acetaldehyde were increased, while BTEX didn't change obviously during a firework episode in Mainz, Germany. The different emission feature of various types of fireworks is a possible explanation of these discrepancies.

The average concentrations of VOCs, CO, $SO_2$ and $NO_x$ at a background period (12:00-18:00 of February $18^{th}$) and a firework episode (F3, 6:00-10:00 of February $19^{th}$) are listed in Table 4, along with their ratios at these two periods. During the firework episode, acetonitrile, aromatics and four OVOCs (acetaldehyde, acetone, acetic acid, and methanol) enhanced obviously. The enhancement ratio for $SO_2$ is the highest, nearly 40. The enhancement ratios for other species are in the range of 2.2-10. For aromatics, these ratios are in the range of 5-10, which agrees well with Chang et al. (2011). For acetaldehyde, acetone, acetic acid, and methanol, these ratios are in the range of 2.2-3.1. In addition, acetonitrile, CO and $NO_x$ enhanced several times as well. The overall enhancement from fireworks indicates that it is a major emission source of gas-phase pollutants during the CNY festival. Hence, more strict control policy on fireworks should be conducted to reduce the concentrations of VOCs and other gas pollutants. From 2005 to 2017, fireworks were allowed in Beijing in specified areas and at specified time. The new policy conducted from 2018 is that fireworks are all prohibited in urban Beijing (within the $5^{th}$ Ring Road), which may reduce the VOCs emission from fireworks. The effectiveness of this new policy in cleaning air during CNY festival needs to be investigated in future studies.

## 4 Conclusions

Wintertime VOCs were online measured by a PTR-MS at suburban and urban Beijing in December 2014 and February 2015, respectively. It is found that the urban VOC concentrations are higher than suburban for all the 14 measured VOCs, with a



factor of 2.67±1.15. However, the VOC concentrations at the suburban site are higher than rural levels, indicating it is influenced by the transportation from urban Beijing and the city groups at Beijing-Tianjin-Hebei area. VOC diurnal variations are different at these two sites, likely due to the stronger local emission (e.g. vehicles) and photochemical processes at the urban site. The emission ratios of VOCs to CO are estimated for both sites, and they are within a factor of

1.5 (except methanol and formic acid), indicating similar anthropogenic emission features. It is also found that the wintertime ERs are much smaller than those at summer, indicating very different emission features at different seasons.

Using emission ratios and CO, we estimated the contributions of primary anthropogenic emission and biogenic/secondary sources for OVOC species. It is found that the photochemical processes play an important role in the secondary formation of OVOCs at the urban site, with the maximum secondary contribution of 50% – 90% during the day. Combining the data of

diurnal variations, ERs, and anthropogenic contributions, we conclude that there are additional sources for methanol at the urban site and formic acid at the suburban site, which are suggested to be solvent use and soil/agriculture, respectively.

The effects of Chinese New Year on VOC characteristics are studied. During CNY holidays, the VOC concentrations decreased ~60%, but the ERs are similar (except for methanol), indicating that the emission intensity decreased drastically because of the population migration, without too much change in the contributions of different anthropogenic emission

sources. The emission ratio of methanol decreased by 86%, verifying its solvent source during the non-holiday period. It is also found that the fireworks are an important source of acetonitrile, aromatics, OVOCs (acetaldehyde, acetone, acetic acid, and methanol), CO, $SO_2$ and $NO_x$ during CNY festival, highlighting the importance of strict emission control of fireworks.

*Author contributions.* MG, KL, and ST designed the research. KL and JL performed the VOCs measurements. RJH provided the CO, $SO_2$ and $NO_x$ data. KL analyzed the data and prepared the manuscript with contributions from all co-authors.

*Competing interests.* The authors declare that they have no conflict of interest.

*Acknowledgements.* This work was supported by the National Key Research and Development Program of China (2016YFC0202202, 2017YFC0212701) and the National Natural Science Foundation of China (Contract No. 91544227, 21477134, 91644219).

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



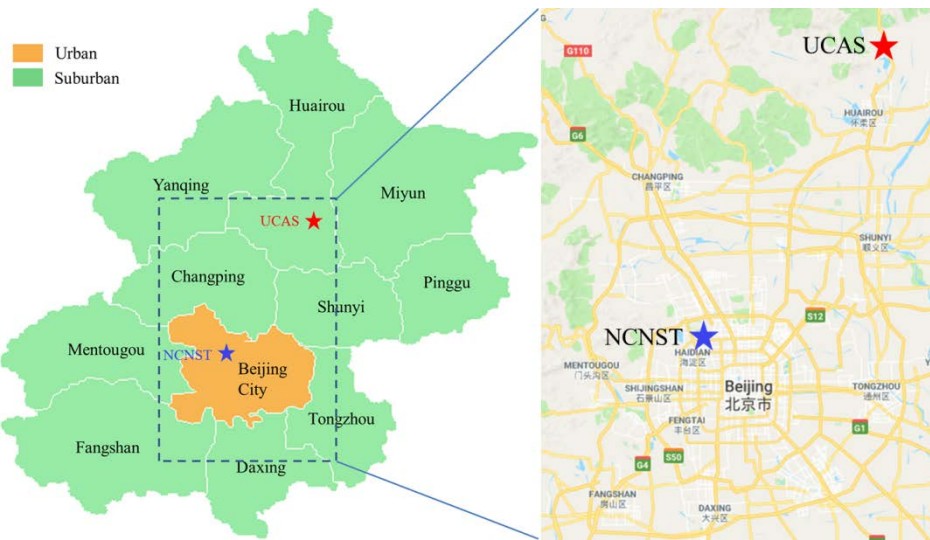

**Figure 1. Locations of the sampling sites. The red star is the suburban site at UCAS; the blue star is the urban site at NCNST (from Google Maps).**

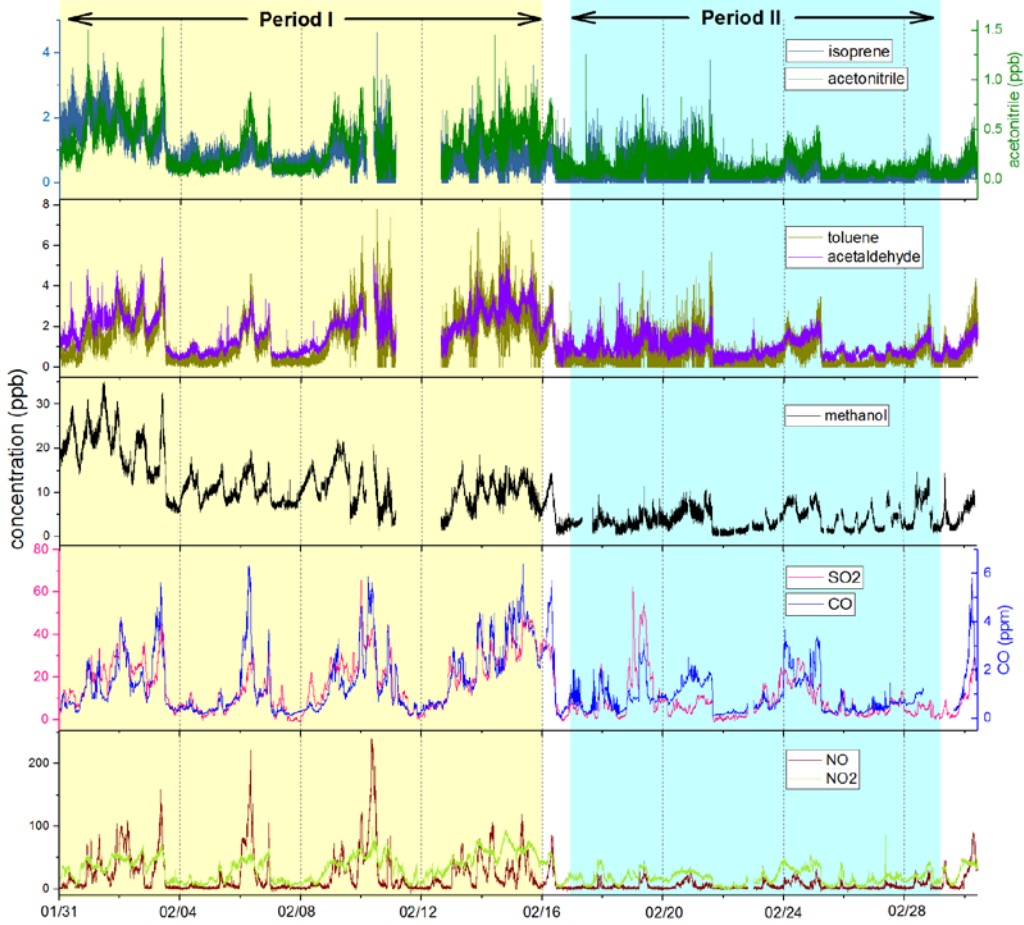





**Figure 2. Time series of representative VOCs, CO, SO$_2$ and NO$_x$ at NCNST urban site. The two periods are shown: Period I is the normal days, while Period II is the CNY holidays.**

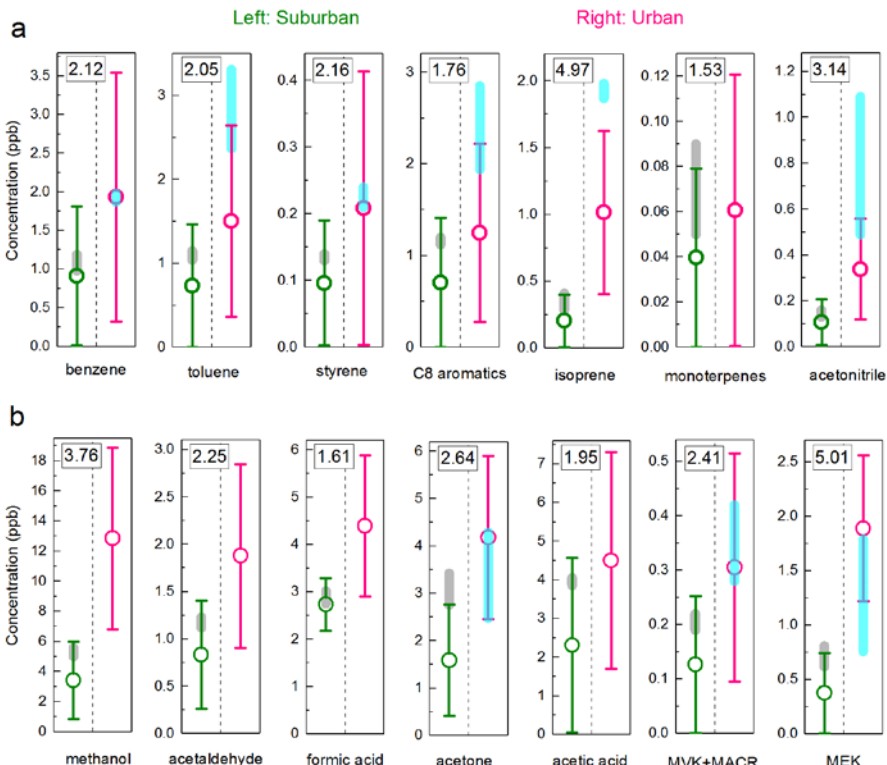

**Figure 3. Suburban and urban VOCs average concentrations: a, hydrocarbons and acetonitrile; b, OVOCs. The circles are VOC concentrations of this work, of which the urban concentration is from the Period I data to avoid any holiday effects. The number at the top is the ratio between urban and suburban average concentrations. The shaded areas are non-APEC average concentrations from previous studies carried out in 2014 autumn at suburban (Li et al., 2017b) and urban (Li et al., 2015) sites.**

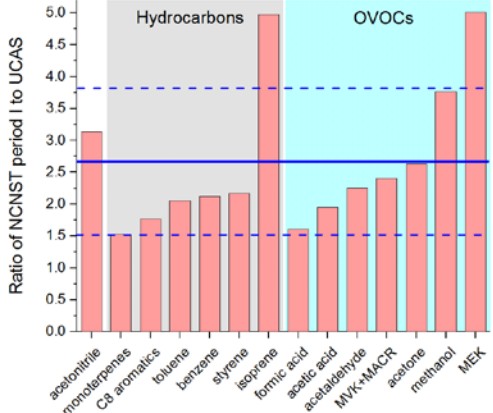

**Figure 4. Ratio of urban site concentration (Period I) to suburban site concentration. The blue lines are average value and standard deviations.**





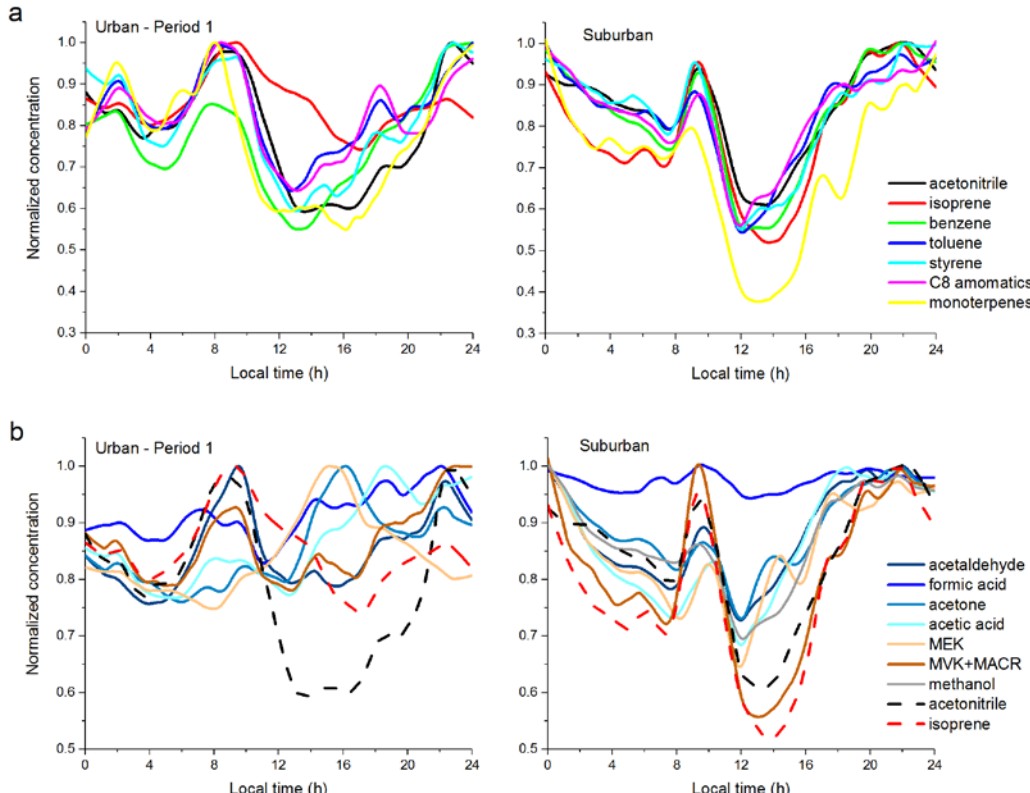

**Figure 5. Diurnal variations of VOCs at urban and suburban sites. a. Hydrocarbons and acetonitrile. b. OVOCs (with acetonitrile and isoprene for comparison).**

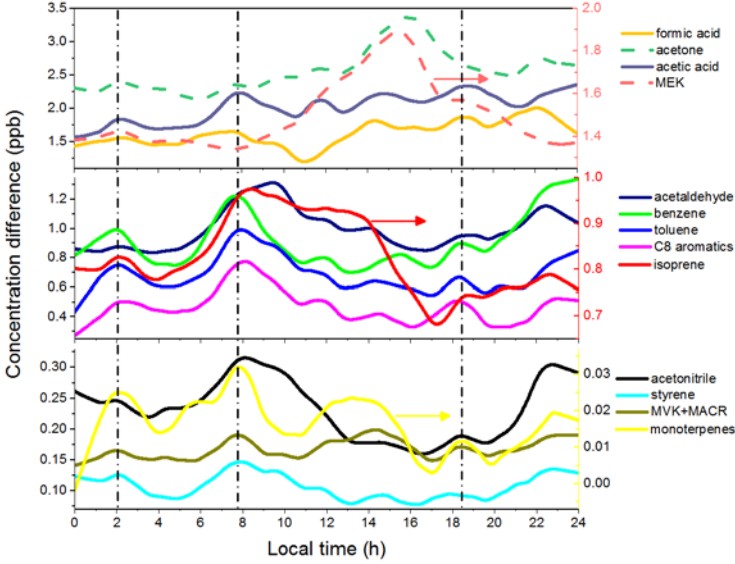





**Figure 6. Diurnal variations of the differences in VOC concentrations at urban and suburban sites. The vertical dashed lines indicate the rush hours related to vehicle emissions.**

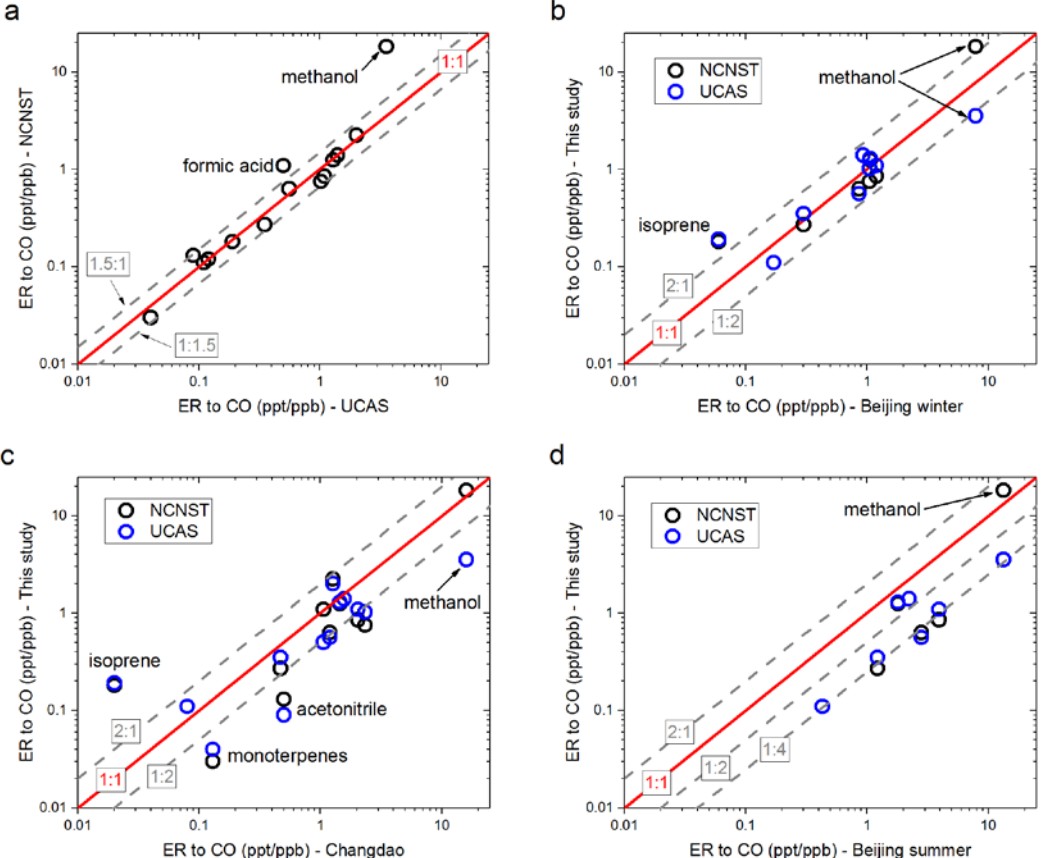

**Figure 7. Comparison of VOC emission ratios relative to CO. a. NCNST site vs UCAS site of this study; b. this study vs urban Beijing winter (Wang et al., 2014); c. this study vs Changdao (Yuan et al., 2013); d. this study vs urban Beijing summer (Yuan et al., 2012).**



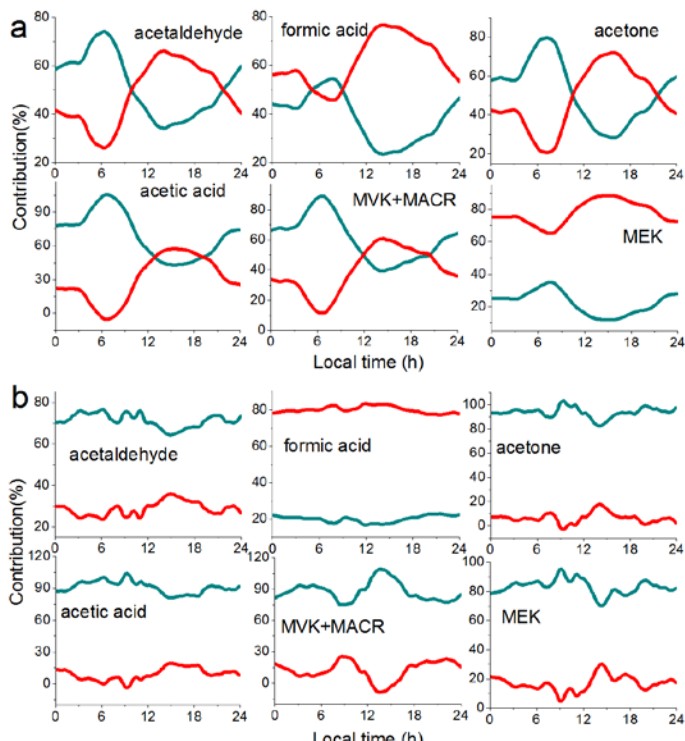

**Figure 8. Contributions of anthropogenic (cyan) and biogenic/secondary (red) sources to OVOC concentrations at the urban site (a) and the suburban site (b).**

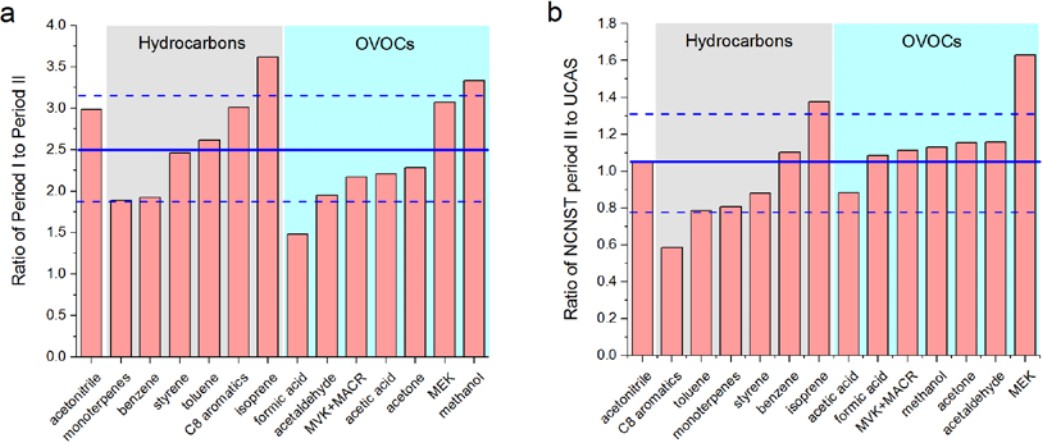

5 **Figure 9. Ratio of: a. concentrations of urban Period I to concentrations of urban Period II; b. concentrations of urban Period II to concentrations of the suburban site. Blue lines are average value of the ratios and standard deviations.**



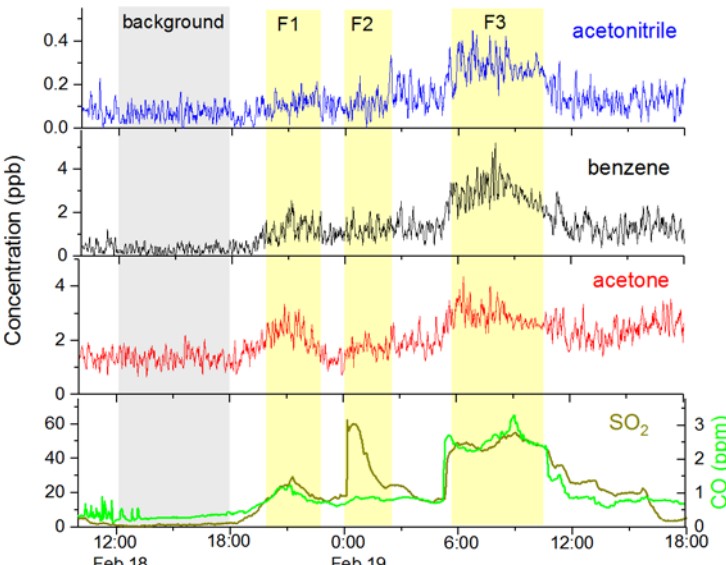

**Figure 10. Time series of acetonitrile, benzene, acetone, SO$_2$, and CO during CNY festival. F1, F2, and F3 represent three firework episodes.**

5    **Table 1. VOC concentrations at urban and suburban sites and comparison to previous studies.**

| Species | This study | | UCAS-suburban (Li et al., 2017b) | PKU-urban (Li et al., 2015) | Changdao (Yuan et al., 2013) |
|---|---|---|---|---|---|
| | UCAS | NCNST-Period I | | | |
| acetonitrile | 0.11±0.10 | 0.34±0.22 | 0.13–0.16 | 0.49–1.09 | 0.21±0.12 |
| acetaldehyde | 0.83±0.57 | 1.88±0.97 | 1.12–1.23 | – | 0.63±0.44 |
| formic acid | 2.73±0.56 | 4.39±1.49 | 2.75–3.02 | – | 2.28±1.02 |
| acetone | 1.59±1.17 | 4.18±1.72 | 2.76–3.42 | 2.48–4.29 | 1.85±0.92 |
| acetic acid | 2.31±2.26 | 4.50±2.80 | 3.86–4.06 | – | 0.77±0.76 |
| isoprene | 0.20±0.20 | 1.01±0.61 | 0.28–0.41 | 0.07–0.11 | 0.01±0.01 |
| MEK | 0.38±0.38 | 1.89±0.67 | 0.62–0.81 | 0.76–1.79 | 0.35±0.22 |
| benzene | 0.91±0.91 | 1.93±1.61 | 0.98–1.19 | 1.87–1.98 | 0.55±0.36 |
| toluene | 0.73±0.73 | 1.51±1.14 | 1.04–1.15 | 2.37–3.31 | 0.57±0.51 |
| styrene | 0.10±0.10 | 0.21±0.21 | 0.13–0.14 | 0.21–0.24 | 0.05±0.04 |
| C8 aromatics | 0.71±0.71 | 1.25±0.97 | 1.15–1.17 | 1.94–2.85 | 0.42±0.39 |
| monoterpene | 0.04±0.04 | 0.06±0.06 | 0.05–0.09 | – | 0.07±0.06 |
| methanol | 3.42±2.58 | 12.86±6.03 | 5.01–5.60 | – | 5.67±4.80 |
| MVK+MACR | 0.13±0.13 | 0.31±0.21 | 0.19–0.22 | 0.28–0.42 | – |



**Table 2. VOC emission ratios to CO at the urban and suburban sites and comparison with other studies.**

| Species | UCAS[a] | R | NCNST Period I[a] | R | Beijing summer[b] | Beijing winter[c] | Changdao[d] |
|---|---|---|---|---|---|---|---|
| acetonitrile | 0.09±0.01 | 0.78 | 0.13±0.01 | 0.76 | -- | -- | 0.5±0.02 |
| acetaldehyde | 0.56±0.03 | 0.83 | 0.63±0.03 | 0.88 | 2.82±0.20 | 0.86±0.04 | 1.2±0.84 |
| formic acid | 0.50±0.06 | 0.64 | 1.09±0.05 | 0.85 | -- | -- | 1.06±0.44 |
| acetone | 1.40±0.10 | 0.78 | 1.40±0.12 | 0.68 | 2.23±0.20 | 0.93±0.04 | 1.57±0.37 |
| acetic acid | 2.01±0.28 | 0.70 | 2.24±0.08 | 0.91 | -- | -- | 1.27±0.32 |
| isoprene | 0.19±0.02 | 0.68 | 0.18±0.04 | 0.24 | -- | 0.06±0.00 | 0.02±0.00 |
| MEK | 0.35±0.02 | 0.75 | 0.27±0.05 | 0.39 | 1.22±0.04 | 0.30±0.02 | 0.47±0.08 |
| benzene | 1.29±0.06 | 0.88 | 1.25±0.04 | 0.92 | 1.80± 0.07 | 1.06±0.03 | 1.45±0.05 |
| toluene | 1.09±0.07 | 0.78 | 0.85±0.03 | 0.89 | 3.93± 0.17 | 1.20±0.05 | 2.05±0.12 |
| styrene | 0.11±0.01 | 0.59 | 0.11±0.01 | 0.67 | 0.43±0.02 | 0.17±0.03 | 0.08±0.01 |
| C8 aromatics | 1.02±0.08 | 0.72 | 0.75±0.03 | 0.87 | -- | 1.05±0.06 | 2.34±0.13 |
| monoterpene | 0.04±0.01 | 0.44 | 0.03±0.00 | 0.62 | -- | -- | 0.13±0.02 |
| methanol | 3.54±0.22 | 0.80 | 18.2±10.3 | 0.16 | 13.4±0.43 | 7.91±0.39 | 16.0±0.67 |
| MVK+MACR | 0.12±0.01 | 0.70 | 0.12±0.01 | 0.81 | -- | -- | -- |

a. This study; b. Yuan et al. (2012); c. Wang et al. (2014); e. Yuan et al. (2013).

**Table 3. Comparison of ERs of the urban site during different periods.**

| Species | Period I | R | Period II | R | change |
|---|---|---|---|---|---|
| acetonitrile | 0.13±0.01 | 0.76 | 0.10±0.01 | 0.77 | -23.1% |
| acetaldehyde | 0.63±0.03 | 0.88 | 0.46±0.03 | 0.82 | -27.0% |
| formic acid | 1.09±0.05 | 0.85 | 0.86±0.06 | 0.75 | -21.1% |
| acetone | 1.40±0.12 | 0.68 | 1.21±0.08 | 0.79 | -13.6% |
| acetic acid | 2.24±0.08 | 0.91 | 1.86±0.11 | 0.80 | -17.0% |
| isoprene | 0.18±0.04 | 0.24 | 0.21±0.03 | 0.54 | 16.7% |
| MEK | 0.27±0.05 | 0.39 | 0.27±0.03 | 0.53 | -0.3% |
| benzene | 1.25±0.04 | 0.92 | 1.36±0.08 | 0.83 | 8.8% |
| toluene | 0.85±0.03 | 0.89 | 0.70±0.05 | 0.76 | -17.6% |
| styrene | 0.11±0.01 | 0.67 | 0.08±0.02 | 0.37 | -27.3% |
| C8 aromatics | 0.75±0.03 | 0.87 | 0.50±0.04 | 0.69 | -33.3% |
| monoterpene | 0.03±0.00 | 0.62 | 0.03±0.01 | 0.31 | -7.4% |
| methanol | 18.2±10.3 | 0.16 | 2.64±0.10 | 0.90 | -85.5% |
| MVK+MACR | 0.12±0.01 | 0.81 | 0.12±0.01 | 0.61 | 4.2% |



**Table 4. Average concentrations of VOCs, CO, SO$_2$, and NO$_x$ at a background period and a firework episode.**

| Species[a] | background | F3 | F3/background |
|---|---|---|---|
| acetonitrile | 0.07±0.07 | 0.29±0.13 | 4.13 |
| benzene | 0.30±0.30 | 3.04±1.05 | 10.01 |
| toluene | 0.24±0.24 | 1.49±0.72 | 6.10 |
| styrene | 0.05±0.05 | 0.27±0.18 | 5.86 |
| C8 aromatics | 0.17±0.17 | 1.15±0.68 | 6.69 |
| acetaldehyde | 0.75±0.30 | 1.74±0.29 | 2.32 |
| acetone | 1.35±0.61 | 2.98±0.77 | 2.21 |
| acetic acid | 1.34±0.77 | 4.10±0.90 | 3.07 |
| methanol | 1.68±0.61 | 4.82±0.87 | 2.87 |
| CO (ppm) | 0.30±0.06 | 2.59±0.27 | 8.70 |
| SO$_2$ | 1.25±0.49 | 49.34±2.91 | 39.55 |
| NO | 2.87±1.43 | 19.44±8.44 | 6.78 |
| NO$_2$ | 9.46±4.40 | 35.47±2.08 | 3.75 |

a. The units of these concentrations are ppb except CO.