# Peer review of "Characteristics of wintertime VOCs in suburban and urban Beijing: concentrations, emission ratios, and festival effects"

_Atmospheric Chemistry and Physics, 2018_

## Referee Comment (RC1) · Anonymous Referee #2 · 28 Mar 2019

**General Comments:**

This manuscript reports the VOC measurements using an Ionicon Q-PTR-MS at both urban and suburban sites of Beijing, China during the wintertime of 2014-2015. Emission ratios (ER) of major VOC species with respect to CO were evaluated and the Chinese New Year effects has been investigated in this study. Wintertime is a typical haze season in China due to the adverse atmospheric dynamic conditions and increasing demands for domestic heating. Therefore, air pollution abatement can be extremely difficult, especially for the Megacities, hosting millions of residents that are vulnerable to air pollutants. VOC have been well recognized to be responsible for the swift development of air pollution events. However, the speciation and emission strength of these VOC have been demonstrated to be hard to acquire due to the fact that VOC can be emitted from a diversity of domestic and industrial activities. Therefore, field measurements of VOC emissions are critically needed in China. This work can be a significant contribution to the atmospheric research community. The methodology (PTR-MS) of this work is well established and the experiments were well conducted. The unit mass resolution of the Q-PTR-MS is the only drawback of this work and I am glad to see that the author has realized this fact and has taken this into account in the data analyses. Overall, the manuscript is fairly well written and I would recommend the manuscript for publication after minor revisions.

Specific Comments:

1) P4, L4: The author may want to specify the operating mode of the PTR-MS, i.e., continue scanning mode or single ion monitor mode.

2) P4, L9: m/z 47 also could be ethanol since most gasoline may contain 10% ethanol in China and it can be emitted into the air from automobile gas tanks. Since no gasstations were around the observation sites, ethanol emission may be not that important though.

3) P4, L11: "Supelco" background check may not the best option for PTR-MS operation. However, it may be OK for the particular dry conditions encountered during wintertime of Beijing.

4) L5, L11: "... the urban site is 3.8 times of that at the suburban site..."

5) P7, L22: "interference of ethanol" very likely.

6) P8, L23-24: "These high...strong photochemical process during the day at the urban site...". The author is most likely right about the photochemical process. In fact, the NOx and VOC levels were substantially higher in urban than in suburban sites. The author may also want to check out the ozone concentrations at both sites, since
ozone is secondary in nature and can well represent the photochemical activity in the atmosphere.

7) P9, L8-9: I think it should be "million".

- 8) P10, L14: remove "were" before "all increased".
- 9) P10, L15: remove "were" before "increased".

10) P10, L16: Change "didn't" into "did not". Change "obviously" into "significantly"

---

## Referee Comment (RC2) · Anonymous Referee #1 · 5 Apr 2019

Review of "Characteristics of wintertime VOCs in suburban and urban Beijing: concentrations, emission ratios, and festival effects" by Li et al.

This manuscript performed VOCs measurements at an urban site and a suburban site in Beijing in winter. The spatial distribution of VOCs is discussed and used to infer the primary and secondary sources of VOCs. The emission ratios are also estimated and contrasted between two sites. It is also shown that the population migration during Chinese new year leads to a 60% decrease in VOCs concentrations. Overall, the conclusions are well supported by the results, though the conclusions are not very exciting. I recommend publication after major revision.

I have some major concerns regarding the discussions on isoprene emission. Firstly, the reported isoprene concentration is about 1ppb at urban site in winter. This concentration is surprisingly high, given the low biogenic isoprene emission in winter. From table 1, isoprene concentration in this study is higher than other studies. Figure 7 also shows that the estimated isoprene emission ratio in this study is higher than other studies. I think it is important to justify the accuracy of isoprene measurement. For example, the authors should better quantify the interference from furan. Secondly, the observation that the daytime reduction of urban isoprene is much lower than other VOCs is intriguing. This phenomenon is most prominent between 12:00 and 16:00. The authors provide three possible explanations on Page 5, but the first two reasons can not explain this observation. It is possible that there is some anthropogenic source of isoprene (after ruling out the interference of furan). If so, this additional unknown source of isoprene would be an important finding. I suggest to look into the sources of isoprene. I want to bring some recent studies on volatile chemical products[1-2] to the authors' attention. Thirdly, in Table 2, the correlation coefficient between isoprene and CO is fairly high, compared to other studies. This also points to a potential anthropogenic source of isoprene. Alternatively, there is substantial interference from furan.

Other comments

1. The VOC vs. CO scatter plot should be shown in the SI for all VOCs.

2. Page 8 Line 16. This conclusion is only applicable to VOC that has very slow reaction rate with oxidants.

Reference

1.      McDonald, B. C.; de Gouw, J. A.; Gilman, J. B.; Jathar, S. H.; Akherati, A.; Cappa, C. D.; Jimenez, J. L.; Lee-Taylor, J.; Hayes, P. L.; McKeen, S. A., et al. Volatile Chemical Products Emerging as Largest Petrochemical Source of Urban Organic Emissions. *Science* **2018,** *359*, 760-764.

2.      Coggon, M. M.; McDonald, B. C.; Vlasenko, A.; Veres, P. R.; Bernard, F.; Koss, A. R.; Yuan, B.; Gilman, J. B.; Peischl, J.; Aikin, K. C., et al. Diurnal Variability and Emission Pattern of Decamethylcyclopentasiloxane (D5) from the Application of Personal Care Products in Two North American Cities. *Environ Sci Technol* **2018,** *52*, 5610-5618.

---

## Author Comment (AC1) · 31 May 2019

**Response to the comments of Anonymous Referee #1**

This manuscript performed VOCs measurements at an urban site and a suburban site in Beijing in winter. The spatial distribution of VOCs is discussed and used to infer the primary and secondary sources of VOCs. The emission ratios are also estimated and contrasted between two sites. It is also shown that the population migration during Chinese new year leads to a 60% decrease in VOCs concentrations. Overall, the conclusions are well supported by the results, though the conclusions are not very exciting. I recommend publication after major revision.

Response: We thank Anonymous Referee #1 for the review and the positive evaluation of our manuscript. We have fully considered the comments and made revisions to our manuscript. The response and changes are listed below.

**Specific Comments:**

I have some major concerns regarding the discussions on isoprene emission.

1) Firstly, the reported isoprene concentration is about 1ppb at urban site in winter. This concentration is surprisingly high, given the low biogenic isoprene emission in winter. From table 1, isoprene concentration in this study is higher than other studies. Figure 7 also shows that the estimated isoprene emission ratio in this study is higher than other studies. I think it is important to justify the accuracy of isoprene measurement. For example, the authors should better quantify the interference from furan.

Response: Thanks for pointing this out. Although m/z 69 from PTR-MS was generally considered to be isoprene ( $C_5H_8H^+$ ) in most previous studies, interferences still remain as indicated by some of them. For example, several previous studies conducted in urban areas (Brito et al., 2015;Valach et al., 2014;Borbon et al., 2013) found that the isoprene emission ratio (ER) obtained from PTR-MS (all ions at m/z 69) was much higher than that measured by GC-based methods, which indicates interferences from other compounds. This might also be one of the reasons why we observed higher isoprene ER than other studies (using GC) in Figure 7.

The interferences at m/z 69 are complex. In addition to furan ( $C_4H_4OH^+$ ) we have mentioned, other interferences may come from fragmentations ( $C_5H_8H^+$ ) of cycloalkanes in urban environments and 2-methyl-3-buten-2-ol (MBO) emitted from pine trees (Yuan et al., 2017;Valach et al., 2014;Kaser et al., 2013). These fragments from cycloalkanes and MBO cannot be distinguished from isoprene even using a PTR-ToF-MS because they are the same ions ( $C_5H_8H^+$ ). Unfortunately, we cannot distinguish any of these compounds (including furan) using the technique in this study, which is one of the limitations of Q-PTR-MS. As a result, the m/z 69 ion is revised to be isoprene+furan+fragments in the new version of manuscript (Figs. 2, 5, and 6).

Here we use two methods in the revised manuscript to better constrain the concentration of isoprene, and these two constraints produce the similar results. The first method is using a fraction of isoprene in m/z 69 to estimate the isoprene concentration. A previous study found that ~22% of the signal at m/z 69 was isoprene in urban London during winter by comparing PTR-MS and GC data, while other signals were mainly cycloalkanes (Valach et al., 2014). When we

apply this fraction to our measurements, the isoprene average concentration (Fig. 3, Table 1) and emission ratios (Fig. 7, Table 2) are very similar to previous studies. Here we only applied this ratio to the statistical data, e.g., average concentration and emission ratios. As the variations of interferences are likely different from isoprene, when we show the time series data (Figs. 2, 5 and 6), the total signal of m/z=69 was used and was marked as isoprene+furan+fragments.

The second method is using MVK+MACR concentration to constrain isoprene concentration, which will be shown in detail in the reply to next comment.

Based on the response above, the following revision was made in the new version of manuscript: Page 4, Line 17-21: "The most significant interferences are at m/z 69, which were previously found mainly from furan, and fragmentations of cycloalkanes in urban environments and 2methyl-3-buten-2-ol (MBO) emitted from pine trees (Kaser et al., 2013;Valach et al., 2014;Yuan et al., 2017). Unfortunately, we cannot distinguish any of these compounds using the technique in this study, hence the m/z 69 ion is considered to be isoprene+furan+fragments."

Page 5, Line 5-9: "As we mentioned in Sect. 2.2, isoprene at m/z 69 may be interfered by furan and fragments from cycloalkanes and MBO. Hence, when the isoprene concentrations are compared with other studies, a factor is applied to the m/z 69 signal. A previous study found that ~22% of the signal at m/z 69 was isoprene in urban London during winter by comparing PTR-MS and gas chromatography (GC) data (Valach et al., 2014). Here we use the same fraction to calculate the isoprene concentrations at both sites, and find that the calculated concentrations are comparable to other studies (Table 1)."

2) Secondly, the observation that the daytime reduction of urban isoprene is much lower than other VOCs is intriguing. This phenomenon is most prominent between 12:00 and 16:00. The authors provide three possible explanations on Page 5, but the first two reasons cannot explain this observation. It is possible that there is some anthropogenic source of isoprene (after ruling out the interference of furan). If so, this additional unknown source of isoprene would be an important finding. I suggest to look into the sources of isoprene. I want to bring some recent studies on volatile chemical products1-2 to the authors' attention.

Reference

1. McDonald, B. C.; de Gouw, J. A.; Gilman, J. B.; Jathar, S. H.; Akherati, A.; Cappa, C. D.; Jimenez, J. L.; Lee-Taylor, J.; Hayes, P. L.; McKeen, S. A., et al. Volatile Chemical Products Emerging as Largest Petrochemical Source of Urban Organic Emissions. *Science* **2018**, *359*, 760-764.

2. Coggon, M. M.; McDonald, B. C.; Vlasenko, A.; Veres, P. R.; Bernard, F.; Koss, A. R.; Yuan, B.; Gilman, J. B.; Peischl, J.; Aikin, K. C., et al. Diurnal Variability and Emission Pattern of Decamethylcyclopentasiloxane (D5) from the Application of Personal Care Products in Two North American Cities. *Environ Sci Technol* **2018**, *52*, 5610-5618.

Response: We agree that the provided references are very important progress in better understanding VOCs emissions; however, they are not directly related to the isoprene sources. Hence, we cited these literatures in the Introduction section of the revised manuscript:

Page 2, Line 6-9: "Urban anthropogenic emissions are complex, for example, recent studies found that the use of volatile chemical products (VCPs) constituted half of fossil fuel VOC emissions in industrialized cities in North America (McDonald et al., 2018;Coggon et al., 2018), however, transportation and industrial emissions are still the main sources in developing countries, e.g., China (Guo et al., 2017)."

Based on our reply to the last comment (only ~22% of the m/z 69 is isoprene) and the following analysis, we think that the lower daytime reduction of m/z 69 in the urban site is mainly from the interferences, rather than from isoprene. **The contents below were added in the SI as Sect. S1.** The concentration of MVK+MACR is used to better constrain the concentration of isoprene. MVK and MACR are mainly from the photooxidation of isoprene, and they can also react with OH (Stroud et al., 2001;Roberts et al., 2006):

| Isoprene + OH $\rightarrow$ 0.23 MACR + 0.33 MVK | $k_1 = 1.0 \times 10^{-10} \text{ cm}^3 \text{ molec}^{-1} \text{ s}^{-1}$ | (S1) |
|--------------------------------------------------|----------------------------------------------------------------------------|------|
| MACR + OH $\rightarrow$ products                 | $k_2 = 2.9 \times 10^{-11} \text{ cm}^3 \text{ molec}^{-1} \text{ s}^{-1}$ | (S2) |
| MVK + OH $\rightarrow$ products                  | $k_3 = 2.0 \times 10^{-11} \text{ cm}^3 \text{ molec}^{-1} \text{ s}^{-1}$ | (S3) |
| Combining these reactions, the MACR/Isopre       | ne ratio can be calculated by:                                             |      |

$$\frac{MACR}{Isoprene} = \frac{0.23k_1}{k_2 - k_1} \left(1 - e^{(k_1 - k_2)[OH]t}\right)$$
(S4)

Using eq. (S4), the isoprene concentration can be constrained.

In this study, only the total concentration of MVK+MACR is measured. Fortunately, the MVK/MACR ratio is generally ~2 during daytime (Stroud et al., 2001). As a result, 1/3 of the total MVK+MACR concentration is considered to be MACR during daytime. Using the MACR concentration and an average OH concentration of  $1 \times 10^6$  molec cm-3 during winter, we estimated the daytime photochemical ages and show them in Figure S4. Three isoprene concentrations were used in this estimation: 1) all m/z 69 signal; 2) m/z 69 multiply a factor of 0.2; 3) m/z 69 signal minus an interference background. As shown in Figure S4, the daytime photochemical age changes are only ~0.2 h and ~0.6 h, when using all m/z 69 signals as isoprene and when using a factor of 0.2, which are highly unlikely. However, the daytime photochemical age change is about 5 h if a background of 0.85 ppb is applied to the m/z 69 signal, which is much more reasonable. Although uncertainties still remain in this method (e.g., using the constant background), it at least indicates that 1) most of the m/z 69 signals are not isoprene, which is similar to Valach et al. (2014); 2) the smaller reduction of m/z 69 during daytime is mainly from interferences. In addition, the interference fraction estimated using this method (84%) is similar with the fraction previous reported (78%) (Valach et al., 2014).

Figure S4. Daytime photochemical age estimated by MACR and isoprene concentrations.

In addition, we rewrote the discussion about isoprene diurnal variation in the main text:

Page 6, Line 13-20: "The daytime reduction of urban m/z 69 (isoprene+furan+fragments) is much lower than other VOCs, and also lower than suburban m/z 69, which may be caused by two reasons. First, the signals at m/z 69 are mainly not from isoprene; instead they are likely from furan and fragments of cycloalkanes (discussed in detail in Sect. S1 of Supporting Information). Furan and cycloalkanes at m/z 69 are mainly from anthropogenic sources such as the combustion and evaporation of fossil fuels (de Gouw and Warneke, 2007;Valach et al., 2014;Yuan et al., 2017). Hence, the emissions of these compounds may be higher in urban areas. Second, this may also be a result of higher isoprene emission at the urban site, as there are some anthropogenic sources of isoprene, e.g., motor vehicles (Borbon et al., 2001;Barletta et al., 2002;Li et al., 2017). This higher urban isoprene emission is further supported by the diurnal variations of MVK+MACR, as discussed below."

3) Thirdly, in Table 2, the correlation coefficient between isoprene and CO is fairly high, compared to other studies. This also points to a potential anthropogenic source of isoprene. Alternatively, there is substantial interference from furan.

Response: In Table 2, the correlation coefficient between isoprene+furan+fragments (m/z 69) and CO is 0.24 at the urban site, which is one of the lowest among all the measured VOCs. The low correlation coefficient is probably because some of the interferences are not co-emit with CO, e.g., cycloalkanes can be emitted from evaporation of gas and oil.

Other comments

1) The VOC vs. CO scatter plot should be shown in the SI for all VOCs. Response: The following figures are added to the SI.

---

## Author Comment (AC2) · 31 May 2019

**Response to the comments of Anonymous Referee #2**

General Comments:

This manuscript reports the VOC measurements using an Ionicon Q-PTR-MS at both urban and suburban sites of Beijing, China during the wintertime of 2014-2015. Emission ratios (ER) of major VOC species with respect to CO were evaluated and the Chinese New Year effects has been investigated in this study. Wintertime is a typical haze season in China due to the adverse atmospheric dynamic conditions and increasing demands for domestic heating. Therefore, air pollution abatement can be extremely difficult, especially for the Megacities, hosting millions of residents that are vulnerable to air pollutants. VOC have been well recognized to be responsible for the swift development of air pollution events. However, the speciation and emission strength of these VOC have been demonstrated to be hard to acquire due to the fact that VOC can be emitted from a diversity of domestic and industrial activities. Therefore, field measurements of VOC emissions are critically needed in China. This work can be a significant contribution to the atmospheric research community. The methodology (PTR-MS) of this work is well established and the experiments were well conducted. The unit mass resolution of the Q-PTR-MS is the only drawback of this work and I am glad to see that the author has realized this fact and has taken this into account in the data analyses. Overall, the manuscript is fairly well written and I would recommend the manuscript for publication after minor revisions.

Response: We thank Anonymous Referee #2 for the review and the positive evaluation of our manuscript. We have fully considered the comments and made revisions to our manuscript. The response and changes are listed below.

**Specific Comments:**

1) P4, L4: The author may want to specify the operating mode of the PTR-MS, i.e., continue scanning mode or single ion monitor mode.

Response: The operating mode of the PTR-MS is multiple ion detection (MID) mode, which is monitoring selected single ions. We added "using the MID (multiple ion detection) mode" in Page 4, Line 6 the revised manuscript.

2) P4, L9: m/z 47 also could be ethanol since most gasoline may contain 10% ethanol in China and it can be emitted into the air from automobile gas tanks. Since no gas-stations were around the observation sites, ethanol emission may be not that important though.

Response: We thank the reviewer for the kind remind. We are aware that ethanol may influence the m/z 47. However, as we have mentioned in the manuscript, the sensitivity of ethanol at m/z 47 is very low (because of fragmentation), and may influence little on formic acid.

We added the possible interferences of formic acid in the method section (Page 4, Line 21-23): "There is also interference at m/z 47, which is mainly from ethanol emitted from solvent or gasoline evaporation. However, the influence of ethanol to formic acid is likely small because the sensitivity of ethanol at m/z 47 is very low as a result of fragmentation (Yuan et al., 2017)." 3) P4, L11: "Supelco" background check may not the best option for PTR-MS operation. However, it may be OK for the particular dry conditions encountered during wintertime of Beijing.

Response: Thanks for pointing this out. The Supelco background check was tested before each campaign by comparing with zero air (generated by AADCO zero air generator), which gave similar background signals. Hence we believe that this is not a big issue.

4) L5, L11: "...the urban site is 3.8 times of that at the suburban site..."

Response: Thanks for the kind remind. In the revised manuscript, "the suburban site is 3.8 times of that at the suburban site" was changed to "the urban site is 3.8 times of that at the suburban site".

5) P7, L22: "interference of ethanol" very likely. Response: Revised in the new version of manuscript.

6) P8, L23-24: "These high...strong photochemical process during the day at the urban site...". The author is most likely right about the photochemical process. In fact, the NOx and VOC levels were substantially higher in urban than in suburban sites. The author may also want to check out the ozone concentrations at both sites, since ozone is secondary in nature and can well represent the photochemical activity in the atmosphere.

Response: Agreed. We added the following statements in Page 9, Line 11-13:

"This can be further proved by the ozone diurnal variations at the urban (Period I only) and suburban sites (Figure S5). The daytime increment of ozone at the urban site (~15 ppb) are ~1.5 times of those at the suburban site (~10 ppb), which indicates stronger daytime photochemical processes at the urban site."

Figure S5 was added in the SI:

Figure S5. Diurnal variations of ozone in urban (Period I only) and suburban sites.

7) P9, L8-9: I think it should be "million".

Response: Yes, "billion" was changed to "million" in the revised manuscript.

8) P10, L14: remove "were" before "all increased". Response: Removed in the revised manuscript.

9) P10, L15: remove "were" before "increased". Response: Removed in the revised manuscript.

10) P10, L16: Change "didn't" into "did not". Change "obviously" into "significantly" Response: We made the corresponding changes in the revised manuscript.